# The Effectiveness of the Use of Regdanvimab (CT-P59) in Addition to Remdesivir in Patients with Severe COVID-19: A Single Center Retrospective Study

**DOI:** 10.3390/tropicalmed7030051

**Published:** 2022-03-18

**Authors:** Ganghee Chae, Aram Choi, Soyeoun Lim, Sooneun Park, Seungjun Lee, Youngick Ahn, Jinhyoung Kim, Seungwon Ra, Yangjin Jegal, Jongjoon Ahn, Eunji Park, Jaebum Jun, Woonjung Kwon, Taehoon Lee

**Affiliations:** 1Department of Internal Medicine, Ulsan University Hospital, University of Ulsan College of Medicine, Ulsan 44033, Korea; margiela07@naver.com (G.C.); 0735483@uuh.ulsan.kr (A.C.); 0733808@uuh.ulsan.kr (J.K.); docra@docra.pe.kr (S.R.); yjjegal@uuh.ulsan.kr (Y.J.); jjahn@uuh.ulsan.kr (J.A.); jjb@uuh.ulsan.kr (J.J.); 2Department of Radiology, Ulsan University Hospital, University of Ulsan College of Medicine, Ulsan 44033, Korea; soyeoun.lim.xr@uuh.ulsan.kr; 3Department of Anesthesiology and Pain Medicine, Ulsan University Hospital, University of Ulsan College of Medicine, Ulsan 44033, Korea; gamju@uuh.ulsan.kr (S.P.); 0735496@uuh.ulsan.kr (S.L.); 0735495@uuh.ulsan.kr (Y.A.); 4Big Data Center, Ulsan University Hospital, University of Ulsan College of Medicine, Ulsan 44033, Korea; 0735779@uuh.ulsan.kr

**Keywords:** regdanvimab, remdesivir, COVID-19, severe

## Abstract

Introduction: Coronavirus disease 2019 (COVID-19) still has a high mortality rate when it is severe. Regdanvimab (CT-P59), a neutralizing monoclonal antibody that has been proven effective against mild to moderate COVID-19, may be effective against severe COVID-19. This study was conducted to determine the effectiveness of the combined use of remdesivir and regdanvimab in patients with severe COVID-19. Methods: From March to early May 2021, 124 patients with severe COVID-19 were admitted to Ulsan University Hospital (Ulsan, Korea) and received oxygen therapy and remdesivir. Among them, 25 were also administered regdanvimab before remdesivir. We retrospectively compared the clinical outcomes between the remdesivir alone group [*n* = 99 (79.8%)] and the regdanvimab/remdesivir group [*n* = 25 (20.2%)]. Results: The oxygen-free days on day 28 (primary outcome) were significantly higher in the regdanvimab/remdesivir group [mean ± SD: 19.36 ± 7.87 vs. 22.72 ± 3.66, *p* = 0.003]. The oxygen-free days was also independently associated with use of regdanvimab in the multivariate analysis, after adjusting for initial pulse oximetric saturation (SpO_2_)/fraction of inspired oxygen (FiO_2_) ratio (severity index). Further, in the regdanvimab/remdesivir group, the lowest SpO_2_/FiO_2_ ratio during treatment was significantly higher (mean ± SD: 237.05 ± 89.68 vs. 295.63 ± 72.74, *p* = 0.003), and the Kaplan-Meier estimates of oxygen supplementation days in surviving patients (on day 28) were significantly shorter [mean ± SD: 8.24 ± 7.43 vs. 5.28 ± 3.66, *p* = 0.024]. Conclusions: In patients with severe COVID-19, clinical outcomes can be improved by administering regdanvimab, in addition to remdesivir.

## 1. Introduction

Coronavirus disease 2019 (COVID-19) is an infectious disease caused by severe acute respiratory syndrome coronavirus 2 (SARS-CoV-2) virus infection [1]. Following the first epidemic that occurred in Wuhan, China in December 2019, there have been 180,654,652 globally confirmed cases until June 2021, of which 3,920,463 have resulted in fatality (case-fatality rate: 2.17%) [2]. In Korea, the first case occurred in January 2020, and by June 2021, 155,572 people were infected, with there being 2015 deaths (case-fatality rate: 1.29%) [2].

Regarding the clinical course of COVID-19 (based on data before remdesivir and systemic corticosteroids were administered) [3], 80% of patients are asymptomatic or have a mild clinical course, 20% develop severe COVID-19 requiring oxygen therapy, and 5% of patients (a quarter of severe COVID-19 cases) progress to critical COVID-19, which requires tracheal intubation or high-flow oxygen therapy, eventually leading to death in 2.3% of all COVID-19 patients [3].

Through large-scale prospective studies conducted in early 2020, it was recognized that the use of remdesivir and systemic corticosteroids was somewhat effective in reducing this mortality rate [4,5]. Accordingly, since September 2020, remdesivir and systemic corticosteroids have been actively used to treat patients with severe COVID-19 in Korea. Korea’s COVID-19 case fatality rate was 1.58% from January 2020 to August 2020 (deaths/cases = 334/21,177) and 1.25% from September 2020 to June 2021 (deaths/cases = 1681/134,395) [2]. As such, it seems that there is a slight reduction in mortality after the use of both drugs. A recent meta-analysis also revealed a decrease in mortality in patients with severe COVID-19 requiring oxygen therapy, after the use of both drugs; however, the degree of mortality reduction was not sufficient [6].

According to the recent Korea Disease Control and Prevention Agency (KDCA) data, which reviewed 8949 COVID-19 patients in Korea, severe COVID-19 with oxygen treatment requirement was observed in 9.1% (816/8949) of cases, of which 29.2% (238/816) resulted in death [7]. Depending on the study, the mortality rate of severe COVID-19 has been reported to be as low as 10% and as high as 36% [3,4,5,7,8,9]. Deaths occur mainly in the elderly (over 60 years of age) and patients with underlying diseases (diabetes mellitus, hypertension, obesity, and chronic heart/kidney/lung disease). In this high-risk group, new drugs other than remdesivir and systemic corticosteroids are urgently needed.

Recently, a monoclonal antibody (mAb) called regdanvimab (CT-P59) was developed by a domestic pharmaceutical company (Celltrion Inc., Incheon, Korea) [10]. Safety and potential antiviral efficacy were confirmed in the phase 1 study [11], and a notable clinical effectivity was confirmed in a phase 2/3 clinical trial, for treatment of mild or moderate COVID-19 patients (reducing hospitalization and oxygen therapy requirement by half, from 8.7% to 4.0%) [12]. It also showed an effect on recently emerged variants [13]. In Korea, regdanvimab has been actively used to treat high-risk patients with mild-to-moderate COVID-19 since March 2021.

Currently, regdanvimab has no clinical usage for treating patients with severe COVID-19, but it has the potential to be effective such cases, when used early in the course of infection. In this study, we evaluated the clinical outcomes of severe COVID-19, when regdanvimab is used in addition to remdesivir and systemic corticosteroids (the current standard of care for severe COVID-19).

## 2. Material and Methods

### 2.1. Study Patients

We retrospectively recruited all severe COVID-19 patients who were admitted and treated with remdesivir and oxygen supplementation at Ulsan University Hospital (UUH) (Ulsan, Korea) from 1 March to 11 May 2021. The recruitment start time was set to 1 March 2021 because regdanvimab has been supplied since the end of February. Since then, regdanvimab has been administered to patients with non-severe COVID-19. Severe COVID-19 patients received remdesivir, and some remdesivir-treated patients had previously been administered regdanvimab because their condition had been non-severe immediately after hospitalization (but progressed to severe COVID-19 during hospitalization). Inclusion criteria and exclusion criteria of the present study were as follows: inclusion criteria (i) those who were diagnosed with COVID-19 and received inpatient treatment at Ulsan University Hospital (UUH) (Ulsan, Korea) from 1 March to 11 May 2021, and (ii) those classified as severe COVID-19 and received remdesivir during the study period; exclusion criteria (i) those who did not received oxygen therapy.

The diagnosis of COVID-19 was made using the real-time polymerase chain reaction (RT-PCR) test for SARS-CoV-2 using the swab sample obtained from the oropharynx and nasopharynx. Severe COVID-19 was defined as the presence of pneumonia and hypoxia [room air pulse oximetric saturation (SpO_2_) ≤ 94%], with laboratory-confirmed SARS-CoV-2 infection. Pneumonia was identified radiologically [via chest X-ray (CXR) or computed tomography] by the presence of an infiltrate. Remdesivir (200 mg IV on the first day, and 100 mg IV from the next day, for a total of 5 days) was administered to patients with symptoms for less than 10 days, among patients with severe COVID-19, according to the KDCA guidelines [14]. Regdanvimab (a single IV dose of 40 mg/kg) was administered to patients with non-severe COVID-19 (room air SpO_2_ > 94%) with laboratory-confirmed SARS-CoV-2 infection, according to the following KDCA guidelines: within 7 days of symptom onset and over 60 years of age, underlying diseases (cardiovascular disease, chronic lung disease, diabetes mellitus, or hypertension), or radiologically identified pneumonia (via CXR or computed tomography) [15].

### 2.2. Study Design

Data were obtained via medical record review from the time of remdesivir initiation to 8 weeks after. After gathering baseline demographic and clinical data at the time of initiation of remdesivir, the following indicators related to clinical outcomes were collected: oxygen use (including the on/off date), type of respiratory support: oxygen with nasal prong or simple mask, or advanced respiratory support [mask with reservoir bag, high flow nasal cannula (HFNC), non-invasive ventilation, invasive ventilation, or extracorporeal membrane oxygenation (ECMO)], SpO_2_/fraction of inspired oxygen (FiO_2_) ratio [16] at the time of initiation of remdesivir, the highest FiO_2_ during treatment, the lowest SpO_2_/FiO_2_ ratio during treatment, CXR scores [17], and survival and hospital discharge.

When a patient was oxygenated via nasal prong, the FiO_2_ values were calculated as 0.24 at 1 L/min, 0.28 at 2 L/min, 0.32 at 3 L/min, 0.36 at 4 L/min, and 0.4 at 5 L/min. When the patient wore a simple oxygen mask, the FiO_2_ value was calculated to be 0.4 for 5–6 L/min. A FiO_2_ value of 0.8 was calculated if the patient was receiving oxygen > 10 L/min with a mask with a reservoir bag [18]. CXR scoring was performed by two thoracic radiologists (W. J. Kwon and S. Lim), with two CXRs at the time around remdesivir start and hospital discharge using the method described in a recent paper [17]. If not discharged until 28 days after starting remdesivir, a CXR around 28 days after remdesivir start was selected as the second point.

After the above investigation, patients were divided into a remdesivir alone group and a regdanvimab/remdesivir group. The outcomes were then analyzed. The present study was approved by the Institutional Review Board of Ulsan University Hospital (IRB number: UUH 2021-06-028).

### 2.3. Primary and Secondary Outcomes

The primary outcome was oxygen-free days on day 28, which was defined as the number of days that a patient was alive and free from oxygen, calculated from the time of initiation of remdesivir. The concept of oxygen-free days is an application of ventilator-free days [19] and has been used in many recent studies [20,21].

As the secondary outcomes, we analyzed the oxygen-free days (on days 14 and 56), days of oxygen supplementation in surviving patients (on days 14, 28, and 56), the highest FiO_2_ and lowest SpO_2_/FiO_2_ ratio during treatment, CXR improvement, duration of hospital stay, and mortality (on days 14, 28, and 56).

### 2.4. Statistical Analysis

Chi-square test and Fisher’s exact test were used to compare the categorical variables. An independent Student’s t-test was used to compare the continuous variables. To identify independent factors associated with oxygen-free days on day 28, multiple linear regression analysis was performed using basic demographic variables [age, sex, and body mass index (BMI)], baseline severity index (SpO_2_/FiO_2_ ratio at the time of initiation of remdesivir), and regdanvimab use. The oxygen supplementation days in surviving patients were estimated using the Kaplan-Meier method and log-rank test. All statistical analyses were performed using SPSS version 24 (IBM Corporation, Armonk, NY, USA). Statistical significance was set at *p* < 0.05.

## 3. Results

### 3.1. Enrolled Patients and Baseline Characteristics

From 1 March to 11 May 2021, 390 symptomatic or high-risk COVID-19 patients were admitted to the UUH. Of these, 74 patients received regdanvimab. Of the 390 patients, 127 were diagnosed with severe COVID-19 and received remdesivir, but three of them did not receive oxygen supplementation at the start of remdesivir administration and were excluded from the analysis. Accordingly, 124 severe COVID-19 patients who received remdesivir and oxygen therapy were included in the present study. Of these, 25 received both regdanvimab and remdesivir (regdanvimab/remdesivir group) and 99 received only remdesivir (remdesivir alone group) (Figure 1).

Table 1 shows the baseline characteristics of the two groups. The mean age was 57.59 years, and 43.5% were male. There was no significant difference in demographic characteristics between the two groups, except for BMI, with that of the regdanvimab/remdesivir group being higher than that of the remdesivir alone group [kg/m^2^, mean ± standard deviation (SD): 24.94 ± 3.19 vs. 26.79 ± 3.83, *p* = 0.014]. Hypertension was the most common underlying disease (25.8%), followed by dyslipidemia (16.1%), diabetes mellitus (12.9%), neurological disease (4.8%), chronic kidney disease (4.0%), and chronic liver disease (4.0%). There was no statistically significant difference in the distribution of underlying diseases between the two groups. The period from symptom onset to hospitalization was slightly shorter in the regdanvimab/remdesivir group (days, mean ± SD: 3.19 ± 3.14 vs. 1.76 ± 3.21, *p* = 0.045), but there was no difference in the period from symptom onset to remdesivir administration (days, mean ± SD: 5.21 ± 3.29 vs. 5.00 ± 3.12, *p* = 0.772). In the regdanvimab/remdesivir group, regdanvimab was administered at 3.68 and 1.92 days, on average, from symptom onset and admission, respectively, and the administration of remdesivir was performed at 1.32 days, on average, after regdanvimab administration. Respiratory support at the start of remdesivir administration showed a tendency to receive less advanced support in the regdanvimab/remdesivir group (10.1% vs. 4.0%, *p* = 0.460), and the SpO_2_/FiO_2_ ratio was significantly higher in the regdanvimab/remdesivir group (290.09 ± 69.76 vs. 326.63 ± 62.39, *p* = 0.018). There was no difference in the baseline CXR severity, and systemic corticosteroids were used in nearly all patients.

### 3.2. Primary Outcome

28 days after the initiation of remdesivir administration, the regdanvimab/remdesivir group showed significantly longer oxygen-free days than the remdesivir alone group (days, mean ± SD: 19.36 ± 7.87 vs. 22.72 ± 3.66, *p* = 0.003) (Table 2).

### 3.3. Secondary Outcomes

In line with the primary outcome, oxygen-free days on days 14 (days, mean ± SD: 7.14 ± 4.26 vs. 8.80 ± 3.43, *p* = 0.074) and 56 (days, mean ± SD: 45.87 ± 12.59 vs. 50.72 ± 3.66, *p* = 0.001) were longer in the regdanvimab/remdesivir group. Survivors for whom oxygen supplementation was stopped also showed a higher tendency in the regdanvimab/remdesivir group on days 14 (80.8% vs. 96.0%, *p* = 0.074), 28 (91.9% vs. 100.0%, *p* = 0.357), and 56 (94.9% vs. 100.0%, *p* = 0.582). The regdanvimab/remdesivir group had a lower FiO_2_ and higher SpO_2_/FiO_2_ ratio during treatment (highest FiO_2_ during treatment, mean ± SD: 0.48 ± 0.26 vs. 0.34 ± 0.15, *p* = 0.001; lowest SpO_2_/FiO_2_ ratio during treatment, mean ± SD: 237.05 ± 89.68 vs. 295.63 ± 72,74, *p* = 0.003). Respiratory support during treatment tended to receive less advanced support in the regdanvimab/remdesivir group (29.3% vs. 12.0%, *p* = 0.077). There were no statistical differences in the degree of CXR change, length of hospitalization, and mortality (Table 2).

According to Kaplan-Meier estimates, the durations of oxygen supplementation in survivors were significantly shorter in the regdanvimab/remdesivir group on days 14 (days, mean ± SD: 6.71 ± 4.18 vs. 5.20 ± 3.43, *p* = 0.046), 28 (days, mean ± SD: 6.71 ± 4.18 vs. 5.20 ± 3.43, *p* = 0.046), and 56 (days, mean ± SD: 9.31 ± 11.41 vs. 5.28 ± 3.66, *p* = 0.024), as compared to the remdesivir alone group (Figure 2).

### 3.4. Independent Factors Associated with Oxygen-Free Days on Day 28

Independent factors associated with oxygen-free days on day 28, identified through multiple linear regression analysis, included regdanvimab use [*B*: 3.568; 95% confidence interval (CI): 0.596–6.539, *p* = 0.019], age (per year, *B*: −0.254; 95% CI: −0.352–−0.156, *p* < 0.001), and baseline SpO_2_/FiO_2_ ratio (*B*: 0.029; 95% CI: 0.013–0.046, *p* = 0.001) (Table 3).

### 3.5. Adverse Events Associated with Regdanvimab Use

Of the 25 regdanvimab users, 14 (56%) had no adverse events. However, some users had the following events: fever in 5 patients (20%), dyspnea in 7 patients (28%), nausea in 1 patient (4%), and delirium in 1 patient (4%). However, these adverse events may be a presentation of COVID-19 itself.

## 4. Discussion

In severe COVID-19 patients, the use of regdanvimab, in addition to remdesivir, increased the number of oxygen-free days. The oxygen-free days was also independently associated with use of regdanvimab, as determined via multiple linear regression analysis with adjustment of baseline severity and demographic variables including age. In addition, the FiO_2_ requirement was lower, the SpO_2_/FiO_2_ ratio was higher, and oxygen dependence was shorter, in patients using regdanvimab. These findings suggest that the use of regdanvimab, in addition to remdesivir, has a significantly favorable impact on the clinical outcomes of severe COVID-19. Our small-scale retrospective study needs to be validated via a prospective, large-scale study. However, given the current high mortality rate of severe COVID-19, the simultaneous use of regdanvimab and remdesivir could be considered in severe COVID-19.

Hospital admissions or deaths due to COVID-19 have been decreasing with the use of vaccines, and vaccines are effective against recently emerged variants [22]. However, once a person develops COVID-19 severe enough to be hospitalized, the mortality rate is very high, ranging from 10–36% [3,4,5,7,8,9]. Although remdesivir and systemic corticosteroids are somewhat effective against severe COVID-19, the associated decrease in mortality rate is insufficient [6]. Deaths from severe COVID-19 occur primarily in people over the age of 60 years and those with underlying medical conditions, and additional new drugs are needed for treating these individuals [3,4,5,7,8,9].

Among several candidate drugs, anti-SARS-CoV-2 mAbs against the receptor-binding domain (RBD) of the spike glycoprotein can be effective in severe COVID-19. This is because they have been proven effective in mild and moderate conditions of the same disease [23,24]. When used early in the course of the disease, such drugs inhibit the progression of mild or moderate COVID-19 to severe COVID-19, which requires hospitalization or oxygen supplementation. As of June 2021, anti-SARS-CoV-2 mAbs that have proven their effectiveness through early phase clinical trials and have secured emergency or conditional use authorization include bamlanivimab (Eli Lily and Company, Indianapolis, IN, USA), a combination of bamlanivimab and etesevimab (Eli Lily and Company, Indianapolis, IN, USA), and a combination of casirivimab and imdevimab (Regeneron Pharmaceuticals, Eastview/Tarrytown, NY, USA) in the United States, as well as regdanvimab (Celltrion Inc., Incheon, Korea) in Korea [23,24].

Regdanvimab is a potent neutralizing antibody against various SARS-CoV-2 isolates, which blocks the interaction regions of the RBD [meant for binding angiotensin-converting enzyme 2 (ACE2)] of SARS-CoV-2 spike protein [10]. Regdanvimab was found to be effective at reducing viral load and ameliorating clinical symptoms in animal experiments [10], and safety and virologic efficacy were confirmed through two randomized phase 1 clinical trials in healthy adults and patients with mild SARS-CoV-2 infection [11]. A recent phase 2/3 clinical trial demonstrated that the progression rates to severe COVID-19 were reduced by 54% (from 8.7% to 4.0%) for patients with mild to moderate COVID-19 and 68% (from 23.7% to 7.5%) for moderate COVID-19 patients aged 50 years and over. Furthermore, the clinical recovery time was 3.4 to 6.4 days faster in patients treated with regdanvimab, compared to those treated with placebo [12]. Recently published retrospective studies also demonstrated that regdanvimab treatment prevented progression to severe disease [25,26]. Accordingly, in February 2021, the Korean Ministry of Food and Drug Safety approved the conditional marketing authorization for the emergency use of regdanvimab for adult (≥18 years) patients with mild to moderate COVID-19, when the following conditions were met: within 7 days of symptom onset and over 60 years of age, underlying disease (cardiovascular disease, chronic lung disease, diabetes mellitus, or hypertension), or radiologically identified pneumonia (either by CXR or computed tomography) [15].

Regdanvimab has been supplied to our hospital (UUH) since the end of February 2021. Since then, regdanvimab has been administered to indicated patients with non-severe COVID-19. Severe COVID-19 patients received remdesivir according to indications, and some remdesivir-treated patients had been treated with regdanvimab because their condition had been non-severe immediately after hospitalization (but progressed to severe COVID-19 during hospitalization). We extracted severe COVID-19 patients who received remdesivir and oxygen therapy after regdanvimab had been supplied and divided them into a group using only remdesevir and a group using both regdanvimab and remdesivir to investigate clinical outcomes.

In our study, the use of regdanvimab in patients with severe COVID-19 significantly increased the number of oxygen-free days, compared to the remdesivir alone group. The use of regdanvimab and increase in oxygen-free days (on day 28) were also significantly associated, as per multiple linear regression analysis that included important clinical indicators such as age, sex, BMI, and baseline SpO_2_/FiO_2_ ratio at the time of initiation of remdesivir administration, which represent the initial severity. Furthermore, the use of regdanvimab was associated with lower FiO_2_ requirement, higher SpO_2_/FiO_2_ ratio during treatment, and a shorter duration of oxygen supplementation. These results suggest that the use of regdanvimab in combination with remdesivir has a significant beneficial effect on the clinical outcomes of severe COVID-19.

Some clinical trials involving targeting of severe COVID-19 by mAbs have already been conducted. In the ACTIV-3 trial (*n* = 326, 1:1 randomization), administration of bamlanivimab, in combination with the standard of care (typically being remdesivir administration), did not demonstrate additional clinical benefits in hospitalized patients with severe COVID-19 [27]. However, in the REGN-COV2 trial (*n* = 9785, 1:1 randomization), a combination of two mAbs (casirivimab and imdevimab), in addition to usual care (not fully disclosed yet), reduced the risk of death by 20%, in patients hospitalized with COVID-19 [28]. The full report of the REGN-COV2 trial has not yet been published, so it is difficult to interpret its results, but the difference between the two conflicting results is presumed to be the timing of mAb injection. In the REGN-COV2 trial, mAbs were administered to early phase patients who were seronegative (before developing an immune response to SARS-COV-2). In the ACTIV-3 trial, which showed negative results, mAb administration took a median of 7 (interquartile range: 5–9) days after symptom onset [27]. In our study, it took an average of 3.68 days for regdanvimab to be administered after symptom onset. For mAbs to be effective in viral infectious diseases, they must be administered in the early stages of infectious diseases, when the viral load is high, but before seroconversion occurs [23]. It is difficult to expect the effect of mAbs in the post-viral phase, wherein a secondary immune response is induced after the initial virally driven phase [23].

There were no serious adverse drug reactions related to the administration of regdanvimab. Fewer than 50% of patients had fever and dyspnea while using regdanvimab, which could be symptoms of COVID-19 itself. None of the users stopped taking the drug due to side effects, and all 25 patients were administered the full dose of regdanvimab without any major events. Also in a previous phase 2/3 trial, there were no reports of serious adverse events associated with the use of regdanvimab [12].

Our study has some important limitations. First, patients with relatively mild COVID-19 might have been included in the regdanvimab/remdesivir group, and patients with relatively severe COVID-19 might have been included in the remdesivir alone group; as the baseline SpO_2_/FiO_2_ ratio was higher in the regdanvimab/remdesivir group, this is a reasonable deduction. However, we found a significant correlation between regdanvimab administration and oxygen-free days, as per the multivariate analysis, adjusted for the SpO_2_/FiO_2_ ratio. And it is also possible that the use of regdanvimab resulted in a relatively mild degree of severe COVID-19. Second, the number of patients was small. Although the results of this study had statistical significance, it is a retrospective, small sample-sized, single-center study; thus, there might be selection bias. Despite these limitations, our study has a novelty in attempting to elucidate the effectiveness of regdanvimab in severe COVID-19.

## 5. Conclusions

Administration of regdanvimab, in addition to remdesivir, significantly improved clinical outcomes in severe COVID-19. Although results of the present study require confirmation via a large-scale, prospective, randomized study, active consideration of regdanvimab administration in severe COVID-19 is needed to facilitate the reduction of high mortality rate associated with severe COVID-19 and the mild adverse drug reaction associated with regdanvimab administration.

## Figures and Tables

**Figure 1 tropicalmed-07-00051-f001:**
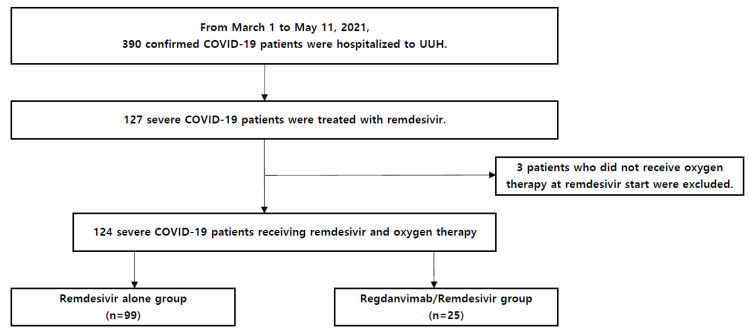
Flowchart of the present study. 390 symptomatic or high-risk COVID-19 patients were hospitalized to UUH from 1 March 1 to 11 May 2021. Of these, 74 patients received regdanvimab. Among 390 hospitalized patients, those receiving remdesivir were selected (*n* = 127). Remdesivir was administered to severe COVID-19 patients with pneumonia (determined via chest imaging) and room air SpO2 ≤ 94%. Three patients were excluded from the study because they did not receive oxygen therapy at the start of remdesivir administration, and finally, 124 severe COVID-19 patients were selected for the current study. Of these, 99 patients were administered only remdesivir, and 25 patients were treated using regdanvimab before remdesivir administration. Regdanvimab was used when room air SpO2 > 94% and chest imaging showed pneumonia, over the age of 60 years, or those with underlying diseases (cardiovascular disease, chronic lung disease, diabetes mellitus, or hypertension). Among the patients who were treated with regdanvimab, remdesivir was administered owing to the change in status to severe COVID-19 during hospitalization. Abbreviations: COVID-19, coronavirus disease 2019; UUH, Ulsan University Hospital; SpO2, pulse oximetric saturation.

**Figure 2 tropicalmed-07-00051-f002:**
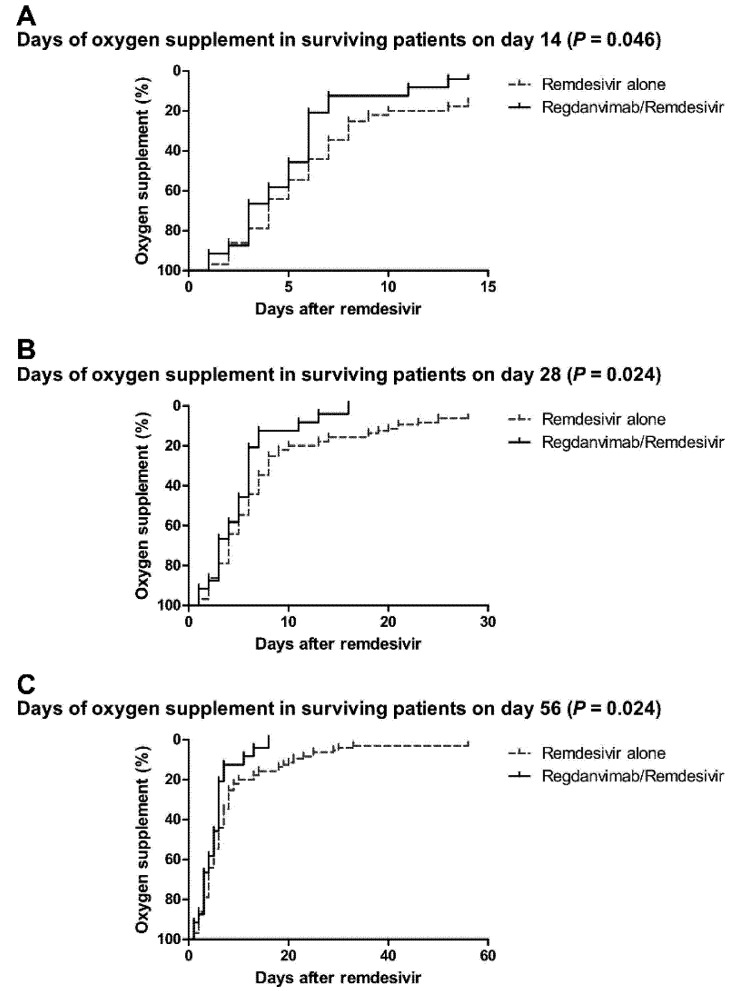
Kaplan-Meier estimates of the remdesivir alone and regdanvimab/remdesivir groups. (**A**) Days of oxygen supplementation, in surviving patients on day 14, were shorter in the regdanvimab/remdesivir group (days, mean ± SD: 6.71 ± 4.18 vs. 5.20 ± 3.43, *p* = 0.046). (**B**) Days of oxygen supplementation, in surviving patients on day 28, were shorter in the regdanvimab/remdesivir group (days, mean ± SD: 8.24 ± 7.43 vs. 5.28 ± 3.66, *p* = 0.024). (**C**) Days of oxygen supplementation, in surviving patients on day 56, were shorter in the regdanvimab/remdesivir group (days, mean ± SD: 9.31 ± 11.41 vs. 5.28 ± 3.66, *p* = 0.024). Abbreviations: SD, standard deviation.

**Table 1 tropicalmed-07-00051-t001:** Baseline characteristics at initiation of remdesivir administration.

Variables		Total (*n* = 124)	Remdesivir Alone (*n* = 99)	Regdanvimab/ Remdesivir (*n* = 25)	*p*-Value
Age (years)		57.59 ± 12.24	56.64 ± 12.13	61.36 ± 12.20	0.085
Age (years), distribution					0.413
	20–29	2 (1.6)	2 (2.0)	0 (0.0)	
	30–39	8 (6.5)	8 (8.1)	0 (0.0)	
	40–49	18 (14.5)	14 (14.1)	4 (16.0)	
	50–59	36 (29.0)	29 (29.3)	7 (28.0)	
	60–69	47 (37.9)	37 (37.4)	10 (40.0)	
	70–79	7 (5.6)	6 (6.1)	1 (4.0)	
	80–89	6 (4.8)	3 (3.0)	3 (12.0)	
Sex					1.000
	Male	54 (43.5)	43 (43.4)	11 (44.0)	
	Female	70 (56.5)	56 (56.6)	14 (56.0)	
Body weight (kg)		68.56 ± 12.68	67.66 ± 11.78	72.12 ± 15.52	0.116
Height (cm)		164.17 ± 8.52	164.35 ± 8.04	163.47 ± 10.34	0.645
BMI (kg/m^2^)		25.31 ± 3.40	24.94 ± 3.19	26.79 ± 3.83	0.014
Race					1.000
	Asian	123 (99.2)	98 (99.0)	25 (100.0)	
	White	1 (0.8)	1 (1.0)	0 (0.0)	
Underlying diseases					
	Hypertension	32 (25.8)	25 (25.3)	7 (28.0)	0.779
	Diabetes	16 (12.9)	12 (12.1)	4 (16.0)	0.738
	Dyslipidemia	20 (16.1)	15 (15.2)	5 (20.0)	0.551
	Chronic heart disease	3 (2.4)	2 (2.0)	1 (4.0)	0.494
	Chronic lung disease	3 (2.4)	2 (2.0)	1 (4.0)	0.494
	Chronic kidney disease	5 (4.0)	4 (4.0)	1 (4.0)	1.000
	Chronic liver disease	5 (4.0)	3 (3.0)	2 (8.0)	0.264
	Rheumatologic disease	2 (1.6)	2 (2.0)	0 (0.0)	1.000
	Neurologic disease	6 (4.8)	5 (5.1)	1 (4.0)	1.000
	Psychiatiric disease	2 (1.6)	2 (2.0)	0 (0.0)	1.000
	Active malignancy	3 (2.4)	2 (2.0)	1 (4.0)	0.494
Days from symptom onset					
	To admission	2.90 ± 3.19	3.19 ± 3.14	1.76 ± 3.21	0.045
	To regdanvimab	NA	NA	3.68 ± 3.00	NA
	To remdesivir	5.17 ± 3.25	5.21 ± 3.29	5.00 ± 3.12	0.772
Days from admission					
	To regdanvimab	NA	NA	1.92 ± 2.08	NA
	To remdesivir	2.27 ± 2.84	2.02 ± 2.89	3.24 ± 2.47	0.055
Days from regdanvimab					
	To remdesivir	NA	NA	1.32 ± 1.77	NA
Respiratory support at the time of initiation of remdesivir					0.460
	Oxygen with nasal prong or simple mask	113 (91.1)	89 (89.9)	24 (96.0)	
	Advanced respiratory support	11 (8.9)	10 (10.1)	1 (4.0)	
	Mask with reservoir bag	2 (1.6)	2 (2.0)	0 (0.0)	
	HFNC	9 (7.3)	8 (8.1)	1 (4.0)	
	NIV	0 (0.0)	0 (0.0)	0 (0.0)	
	Invasive ventilation	0 (0.0)	0 (0.0)	0 (0.0)	
	ECMO	0 (0.0)	0 (0.0)	0 (0.0)	
FiO_2_ at the time of initiation of remdesivir		0.34 ± 013	0.35 ± 0.14	0.30 ± 0.08	0.078
SpO_2_/FiO_2_ ratio at the time of initiation of remdesivir		297.46 ± 69.67	290.09 ± 69.76	326.63 ± 62.39	0.018
SpO_2_/FiO_2_ ratio distribution at the time of initiation of remdesivir					0.213
	0–99	3 (2.4)	3 (3.0)	0 (0.0)	
	100–199	8 (6.5)	7 (7.1)	1 (4.0)	
	200–299	39 (31.5)	33 (33.3)	6 (24.0)	
	300–399	69 (55.6)	54 (54.5)	15 (60.0)	
	400–499	5 (4.0)	2 (2.0)	3 (12.0)	
CXR score at the time of initiation of remdesivir		5.16 ± 4.31	5.42 ± 4.42	4.16 ± 3.75	0.194
Systemic corticosteroids use		122 (98.4)	97 (98.0)	25 (100.0)	1.000
	Dexamethasone	121 (97.6)	97 (98)	24 (96.0)	
	Prednisolone	1 (0.8)	0 (0.0)	1 (4.0)	

Data are presented as mean ± standard deviation or number (%). BMI: body mass index; CXR: chest X-ray; NA: not applicable; HFNC: high flow nasal cannula; NIV: non-invasive ventilation; ECMO: extracorporeal membrane oxygenation; FiO2: fraction of inspired oxygen; SpO2: pulse oximetric saturation.

**Table 2 tropicalmed-07-00051-t002:** Clinical outcomes of the remdesivir alone group and the regdanvimab/remdesivir group.

Variables	Total (*n* = 124)	Remdesivir Alone (*n* = 99)	Regdanvimab/ Remdesivir (*n* = 25)	*p*-Value
Primary outcome				
Oxygen-free days on day 28				0.003
Mean ± SD	20.04 ± 7.33	19.36 ± 7.87	22.72 ± 3.66	
Median (IQR)	22.0 (20.0–24.5)	22.0 (19.0–24.0)	23.0 (22.0–25.0)	
Secondary outcomes				
Oxygen-free days on day 14				0.074
Mean ± SD	7.48 ± 4.15	7.14 ± 4.26	8.80 ± 3.43	
Median (IQR)	8.0 (6.0–10.5)	8.0 (5.0–10.0)	9.0 (8.0–11.0)	
Oxygen-free days on day 56				0.001
Mean ± SD	46.85 ± 11.52	45.87 ± 12.59	50.72 ± 3.66	
Median (IQR)	50.0 (48.0–52.5)	50.0 (47.0–52.0)	51.0 (50.0–53.0)	
Oxygen off and live on day 14	104 (83.9)	80 (80.8)	24 (96.0)	0.074
Oxygen off and live on day 28	116 (93.5)	91 (91.9)	25 (100.0)	0.357
Oxygen off and live on day 56	119 (96.0)	94 (94.9)	25 (100.0)	0.582
The highest FiO_2_ during treatment	0.45 ± 0.24	0.48 ± 0.26	0.34 ± 0.15	0.001
The lowest SpO_2_/FiO_2_ ratio during treatment	248.86 ± 89.43	237.05 ± 89.68	295.63 ± 72.74	0.003
The lowest SpO_2_/FiO_2_ ratio distribution during treatment				0.087
0–99	15 (12.1)	15 (15.2)	0 (0.0)	
100–199	17 (13.7)	14 (14.1)	3 (12.0)	
200–299	41 (33.1)	34 (34.3)	7 (28.0)	
300–399	50 (40.3)	35 (35.4)	15 (60.0)	
400–499	1 (0.8)	1 (1.0)	0 (0.0)	
The highest degree of respiratory support during treatment				0.077
Oxygen with nasal prong or simple mask	92 (74.2)	70 (70.7)	22 (88.0)	
Advanced respiratory support	32 (25.8)	29 (29.3)	3 (12.0)	
Mask with Reservoir bag	0 (0.0)	0 (0.0)	0 (0.0)	
HFNC	20 (16.1)	18 (18.2)	2 (8.0)	
NIV	0 (0.0)	0 (0.0)	0 (0.0)	
Invasive ventilation	12 (9.7)	11 (11.1)	1 (4.0)	
ECMO	0 (0.0)	0 (0.0)	0 (0.0)	
Changes in CXR				
Days from the first scored CXR	10.93 ± 6.69	11.45 ± 7.03	8.96 ± 4.80	0.098
Difference between the two CXR scores (initial minus post)	1.36 ± 4.66	1.58 ± 4.59	0.48 ± 4.95	0.294
Duration of hospital stay (days)	15.40 ± 10.38	15.67 ± 11.12	14.32 ± 6.78	0.563
Mortality *				
Death at day 14	1 (0.8)	1 (1.0)	0 (0.0)	1.000
Death at day 28	2 (1.6)	2 (2.0)	0 (0.0)	1.000
Death at day 56	2 (1.6)	2 (2.0)	0 (0.0)	1.000
All-cause mortality	2 (1.6)	2 (2.0)	0 (0.0)	1.000

Data are presented as mean ± standard deviation or number (%). SD: standard deviation; IQR: interquartile range; FiO_2_: fraction of inspired oxygen; SpO_2_: pulse oximetric saturation; HFNC: high flow nasal cannula; NIV: non-invasive ventilation; ECMO: extracorporeal membrane oxygenation; CXR: chest X-ray. * Two deaths occurred: a 78-year-old male and an 83-year-old female died on days 15 and 12, respectively. Their cause of death was COVID-19.

**Table 3 tropicalmed-07-00051-t003:** Factors associated with oxygen-free days on day 28.

Variables	Simple Linear Regression							Multiple Linear Regression						
	*B*	SE	β	t	*p*-Value	95% CI for B		*B*	SE	β	t	*p*-Value	95% CI for B	
						Lower	Upper						Lower	Upper
Regdanvimab use	3.356	1.619	0.184	2.073	0.040	0.151	6.562	3.568	1.501	0.196	2.377	0.019	0.596	6.539
Age (per year)	−0.246	0.049	−0.411	−4.981	<0.001	−0.344	−0.148	−0.254	0.049	−0.424	−5.126	<0.001	−0.352	−0.156
Female sex	2.761	1.309	0.188	2.109	0.037	0.169	5.353	1.742	1.140	0.118	1.528	0.129	−0.516	4.000
BMI (per kg/m^2^)	0.306	0.193	0.142	1.585	0.116	−0.076	0.689	−0.043	0.181	−0.020	−0.238	0.812	−0.402	0.316
Baseline SpO_2_/FiO_2_ ratio	0.040	0.009	0.383	4.582	<0.001	0.023	0.058	0.029	0.009	0.280	3.461	0.001	0.013	0.046

CI: Confidence Interval; SE: standard error; BMI: body mass index; SpO_2_: pulse oximetric saturation; FiO_2_: fraction of inspired oxygen, SpO_2_: pulse oximetric saturation.

## Data Availability

No new data were created or analyzed in this study. Data sharing is not applicable to this article.

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
