# Peer review of "The Effectiveness of the Use of Regdanvimab (CT-P59) in Addition to Remdesivir in Patients with Severe COVID-19: A Single Center Retrospective Study"

_tropicalmed, 2022, doi:10.3390/tropicalmed7030051_

Round 1
Reviewer 1 Report
Dear Authors, the data presented is remarkably interesting and informative towards the scientific community in this tough time. I have a few more concerns bellow -
- It will be better to make clear age groups and how many people belong to this group.
- There are some studies from Korea itself regarding the uses of regdanvimab and remedisivir. In the introduction and discussion section, it will be better to describe a bit and link your study with proper citation.
- How the infection was confirmed by RT-PCR or Antigen test! Also, it's important to mention how the sample was collected. Those parts are important to mention from a biological relevance point of view.
- Also, you should mention why your study is significant because there are few more studies from Korea itself regarding this MS topic.
Reviewer 2 Report
The authors conducted a retrospective cohort study with the aim to ascertain the difference in prognosis between the remdesivir alone group and the remdesivir/regdanvimab. The authors are applauded for the novelty of investigating the effectiveness of remdesivir/regdanvimab. However, there are two major concerns; case selection criteria are not clear and patients in the remdesivir/regdanvimab group were administered regdanvimab because of the mild clinical course then were administered remdesivir due to exacerbation. Therefore, it is not convincing that the combined use of remdesivir and regdanvimab is more effective than remdesivir alone, unless more analyses are done to justify.
Please also find below issues.
Study patients:
- Were all those admitted to UUH from 1st March 11th May 2021 included in the very first step of case selection? Or there was exclusion criterion?
- This section is confusing. My understanding is that 390 patients were admitted to UUH during the study time. Of these 127 were administered remdesivir and were severe, defined as pneumonia and hypoxia. Then 3 were excluded because oxygen therapy was not implemented. For the remaining 124 patients, 99 had remdesivir alone and 25 had both remdesivir and regdanvimab. Is this correct?
- The authors need to state explicitly the inclusion and exclusion criteria, preferably in a point form such as (i), (ii), etc.
- If a severe case was defined by pneumonia and hypoxia, why were the 3 patients excluded due to oxygen therapy not needed? Did you mean they were excluded due to hypoxia? Labelling in Figure 1 should match the inclusion and exclusion criteria.
- Should the box “124 severe COVID-19 patients …” in Figure 1 also include the pneumonia condition?
- The sentence “Among them, 74 patients were treated with regdanvimab” should be removed in Figure 1. It is an event of the study, not an inclusion or exclusion criterion.
- Also Figure 1 should be in the Results section. The Method section aims to describe the procedure as if the study has not been conducted.
- Line 89-90: So some patients had regdanvimab (because it was mild) then were administered remdesivir for the exacerbation.
- Line 106: How was pneumonia defined? Did you exclude bacterial pneumonia due to secondary infection?
Statistical analysis
- Did you include succumbed patients in the statistical analysis?
- If so, were they succumbed to SARS-CoV-2 infection?
- Because they are time-to-event data, survival analysis such as Cox regression should be used instead of linear regression.
Discussion
- “Fewer than 50% of patients 323 had fever and dyspnea while using regdanvimab (data not shown)” This is concerning and the authors should provide the information of possible adverse effect of regdanvimab in the Result section.
- Also, “the additional use of regdanvimab in combination with remdesivir” is rather misleading. As addressed, remdesivir was administered due to exacerbation.
Round 2
Reviewer 2 Report
None.